# From Simple to Complex: A Progressive Framework for Document-level Informative Argument Extraction

**Quzhe Huang[1,2]**[*] **Yanxi Zhang[1,3]**[*] **✉ Dongyan Zhao[1,4]**
[1]Wangxuan Institute of Computer Technology, Peking University
[2]School of Intelligence Science and Technology, Peking University
[3]Center for Data Science, Peking University
[4]National Key Laboratory of General Artificial Intelligence
{huangquzhe,zhaody}@pku.edu.cn
zhangyx@stu.pku.edu.cn

## Abstract

Document-level Event Argument Extraction (EAE) requires the model to extract arguments of multiple events from a single document. Considering the underlying dependencies between these events, recent efforts leverage the idea of "memory", where the results of already predicted events are cached and can be retrieved to help the prediction of upcoming events. These methods extract events according to their appearance order in the document, however, the event that appears in the first sentence does not mean that it is the easiest to extract. Existing methods might introduce noise to the extraction of upcoming events if they rely on an incorrect prediction of previous events. In order to provide more reliable memory, we propose a simple-to-complex progressive framework for document-level EAE. Specifically, we first calculate the difficulty of each event and then, we conduct the extraction following a simple-to-complex order. In this way, the memory will store the most certain results, and the model could use these reliable sources to help the prediction of more difficult events. Experiments on WIKIEVENTS show that our model outperforms SOTA by 1.4% in F1, indicating the proposed simple-to-complex framework is useful in the EAE task. The code is available at https://github.com/zhangyx0417/simple_to_complex.

## 1 Introduction

Document-level Event Argument Extraction (EAE) aims at identifying the participants of multiple events from a document and classifying them into proper roles (Li et al., 2021; Du et al., 2022; Xu et al., 2022; Huang et al., 2022; Yang et al., 2023). Understanding events in documents is crucial for a line of downstream tasks, such as machine reading comprehension (Han et al., 2021) and dialogue systems (Zhang et al., 2020).

Generation-based document-level EAE methods are widely used in recent works (Li et al., 2021; Du et al., 2022; Du and Ji, 2022; Huang et al., 2022). Among them, one line of studies (Li et al., 2021; Huang et al., 2022) treats each event independently and ignores the underlying correlations between events in real-world documents. Other works (Du et al., 2022; Du and Ji, 2022) start to consider inter-event dependencies and model them by introducing the idea of "memory", where event predictions (e.g., arguments, roles) are cached and can be retrieved to help the prediction of the upcoming events in a document. However, since these methods still use *front-to-back* prediction—extracting events according to their appearance order in a document, an event can only rely on the predictions of events that appeared before it. Besides, the prediction of an event is cached regardless of its quality, whereas false predictions may be cached first, misleading the prediction of the following events.

In general, using current retrieval-based methods to model inter-event dependencies faces two main challenges: (1) *front-to-back* prediction only partially models inter-event dependencies, where the dependency links from an event to all the events that appeared after it are ignored; (2) incorrect predictions may be cached first and retrieved by the upcoming events.

Considering the challenges, we propose the *simple-to-complex* framework. First, we calculate the difficulty of each event, where the difficulty is defined as the average probability that the model assigns to the arguments of an event. We also calibrate the argument probabilities before using them to ensure they truly reflect how certain the model is on each argument. Second, we reorder events in a document from simple to complex and predict them accordingly. In this way, the model could use the simple instance to help the prediction of the difficult ones, no matter whether the simple events appear before or after the difficult ones in

---

[*] Equal Contribution.

the original document.

We conduct experiments on widely used benchmarks WIKIEVENTS(Li et al., 2021), and our proposed simple-to-complex framework outperforms the previous SOTA by 1.4% in F1, illustrating the effectiveness of our method. Further analyses show the calibration of probability is very important when calculating the difficulty of different events and the success of our framework relies on the better usage of dependency between an event and the events that appear after it in the document.

## 2 Task Definition

In this work, we focus on document-level **Informative Argument Extraction**[1] (IAE) (Li et al., 2021), where informative arguments are far more distant than local/uninformative ones and provide more useful information about an event. We formulate document-level IAE as a generative template-filling task following Li et al. (2021) and Du et al. (2022). Given a document $D$ with triggers marked (using a special token <tgr>), our goal is to extract all the arguments of $E$ to fill in the slots of the event template $T$.

Each event consists of (1) an **event trigger**, which has a specific type $E$ (we use $E$ to represent an event); (2) a series of **event arguments**, each corresponding to a specific role. In the event ontology, event types and argument roles are pre-defined, and event templates depicting the relationship between the argument roles of an event are also provided. For example, the template for $E = Attack$ in the KAIROS ontology[2] is:

<arg1> attacked <arg2> using

<arg3> at <arg4> place

where each slot <argx> is a placeholder for arguments with a specific role. We replace all the <argx>s in a template with a special token <arg> before using them as input. If the model extracts an argument, then <arg> will be replaced by the argument. If no, the placeholder <arg> remains.

## 3 Method

In this section, we illustrate our framework based on simple-to-complex prediction (Figure 1). First, we introduce our memory-enhanced IAE model

[1]Name mentions are more informative than nominal mentions, and pronouns are the least informative.

[2]https://www.ldc.upenn.edu/collaborations/current-projects

(Section 3.1). Here, we use retrieval to augment model input and apply constrained decoding to improve model output, both leveraging inter-event dependencies to benefit prediction. Second, we elaborate on how to define and calculate the difficulty of an event, and how to reorder events in a document from simple to complex for simple-to-complex prediction (Section 3.2).

### 3.1 Memory-Enhanced IAE Model

Our memory-enhanced IAE model is based on a generative model. When calculating the difficulty of each event as well as generating (the arguments of) events in a document from simple to complex, we use the same generative model. In this study, we assume that each event has a prediction order and events in a document are predicted according to that order. After reordering, the prediction order of an event may change and further change the retrieved prediction of that event.

**Model Input & Output**   The generation of an event $E$ in a document $D$ is conditioned on the (1) **prediction order** $o \in \{1, 2, \ldots, n_e\}$: the order of predicting (the arguments of) $E$, where $n_e$ denotes the number of events in $D$; (2) **event context** $c$: the concatenation of $E$'s *context words* (a continuous span in $D$ close to $E$'s trigger) and $E$'s template; (3) **retrieved prediction** $m^R$: the prediction of an event appeared before $E$ retrieved from the *document memory*, a data structure that caches the predictions of already predicted events in $D$. To sum up, the input of event $E$ for the model is:

 $m^R$   $T$  $x_1, x_2, \ldots, x_n$ [EOS]

where $x_1, x_2, \ldots, x_n$ are the context words and $T$ is $E$'s unfilled template, and these two parts form the event context $c$. The prediction of $E$ is a filled template, with each <arg> placeholder replaced by the predicted argument (or not), as shown in Figure 1. If there are multiple arguments for the same slot, the arguments are connected with "and".

**Retrieval-Augmented Generation**   In the input stage (both for training and testing), we augment our model with similarity-based retrieval following Du et al. (2022) to make it capable of finding argument mentions beyond the context of an event, especially informative ones (Li et al., 2021). When predicting the $i$-th event $E_i$ in a document $D$, the snapshot of the document memory is $\mathbf{m} = \{m_1, m_2, \ldots, m_{i-1}\}$, where $m_k$ denotes the prediction of the $k$-th event. We calculate the cosine

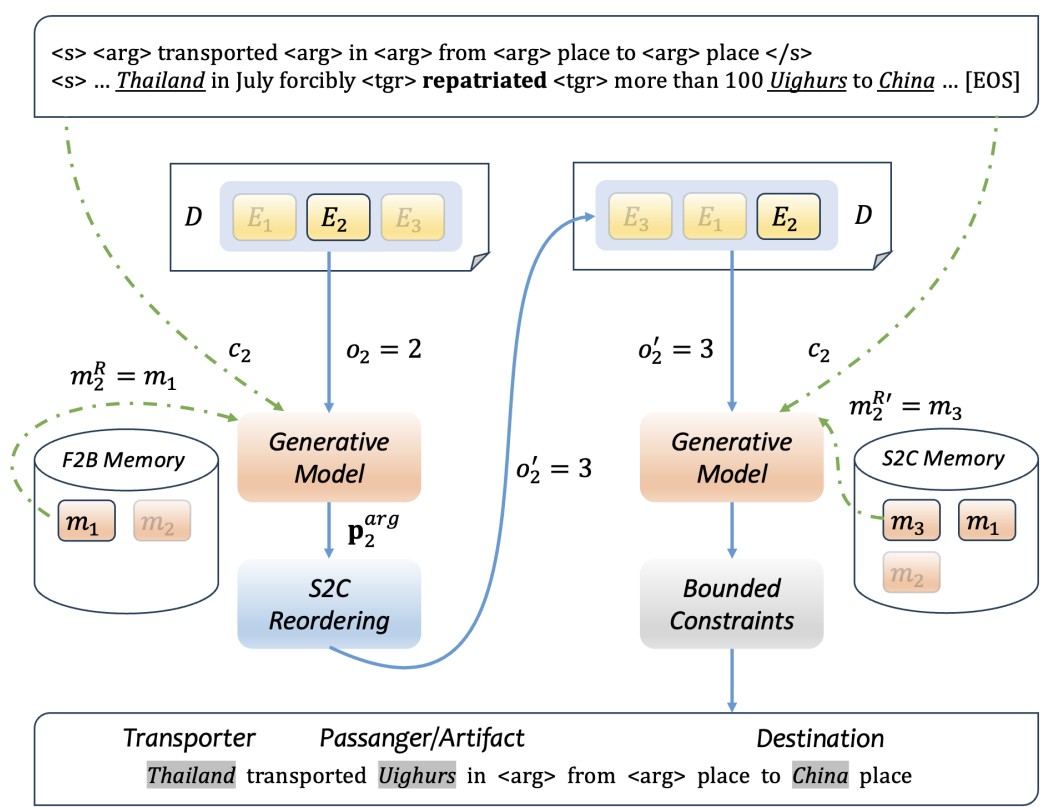

Figure 1: Our simple-to-complex progressive framework for document-level IAE. First, we calculate the difficulty of each event in a document $D$ and obtain a new prediction order for that event. Second, we reorder events in $D$ from simple to complex, and predict them accordingly. S2C denotes Simple-to-Complex, while F2B denotes Front-to-Back. Here, we plot the process of predicting the arguments of $E_2$.

similarity between $E_i$'s context $c_i$ and each prediction in $\mathbf{m}$ using S-BERT (Reimers and Gurevych, 2019) embeddings, and select the prediction with the highest score as additional input to help the prediction of $E_i$:

$$\text{score}(m_j|c_i) = \frac{\exp f(c_i, m_j)}{\sum_{m_j \in m} \exp f(c_i, m_j)} \quad (1)$$

$$f(c_i, m_j) = \text{SBERT}(c_i)^T \text{SBERT}(m_j) \quad (2)$$

$$m_i^R = \arg\max_{m_j} \text{score}(m_j|c_i) \quad (3)$$

where $\text{SBERT}()$ denotes S-BERT encoding, $m_i^R$ denotes the retrieved prediction that $E_i$ relies on.

**Constrained Decoding** In the output stage, we introduce argument pair constraints following Du et al. (2022) to constrain the decoding of arguments with conflicting roles. For example, if the DE-TAINEE of event $E_a$ is "Mike", then "Mike" can not be decoded as the ATTACKER of another event $E_b$ (happened after $E_a$), since "Mike" is already in jail. Here, "Mike" as DETAINEE and "Mike" as AT-TACKER is an argument pair constraint. However, once an incorrect prediction is used to constrain

another, it may cause more errors (Du et al., 2022). In the example above, "Mike" will never be decoded as the ATTACKER of $E_b$ once it is decoded as the DETAINEE of $E_a$, even if the prediction of DETAINEE is incorrect. To alleviate such error propagation, we disable the constraints when the model is not certain about the prediction of an argument. The *certainty* of an argument can be measured by the calibrated probability of decoding it. Low probability intuitively implies the model is not confident about this prediction, while we have also found that high probability (e.g. $\geq 0.8$) corresponds to low prediction accuracy, which is shown in Figure 3. Therefore, we set both lower and upper bounds for argument probabilities to exclude possibly incorrect constraints, and we refer to our pruned constraints as **bounded constraints**. The heuristics of selecting bounds are discussed in Appendix A.3.

### 3.2 Simple-to-Complex Reordering

Since an event usually contains multiple arguments, we reckon the difficulty of an event lies in the average difficulty of predicting its arguments. In this

section, we will elaborate on how to calculate the difficulty of an event as well as how to reorder events in a document by the difficulty and predict them from simple to complex.

Suppose the set of prediction orders of events in a document $D$ be $\mathbf{o} = \{o_1, o_2, \ldots, o_{n_e}\}$, where $o_i$ represents the $i$-th appeared event is the $o_i$-th to be predicted, then front-to-back prediction satisfies $o_i = i, i = 1, 2, \ldots, n_e$. Suppose the probabilities of decoding the arguments of the $i$-th event $E_i$ in a document $D$ be:

$$\mathbf{p}_i^{arg} = \left\{ p_i^{(1)}, p_i^{(2)}, \ldots, p_i^{(n_a)} \right\}, \qquad (4)$$

where $n_a$ denotes the number of predicted arguments of $E_i$, and $p_i^{(j)}$ denotes the probability that the generative model assigns to the $j$-th argument of the $i$-th event. The argument probability reflects the certainty of the generative model on predicting an argument, inversely proportional to our desired difficulty. Thus, we define the difficulty of predicting the arguments of $E_i$ as:

$$\mathbf{d}_i^{arg} = \left\{ d_i^{(1)}, d_i^{(2)}, \ldots, d_i^{(n_a)} \right\}, \qquad (5)$$

$$d_i^{(j)} = 1 - p_i^{(j)}, j = 1, 2, \ldots, n_e, \qquad (6)$$

where $d_i^{(j)}$ denotes the difficulty of predicting the $j$-th argument of the $i$-th event. The difficulty of $E_i$ is defined as the average difficulty of predicting its arguments, so we take the average of $\mathbf{d}_i^{arg}$ and obtain the difficulty of $E_i$:

$$d_i^{evt} = \mathrm{mean}(\mathbf{d}_i^{arg}). \qquad (7)$$

If no arguments of $E_i$ are predicted, then $d_i^{evt} = 2$. That means $E_i$ will be placed to the rear, since it provides no arguments/roles that might benefit prediction. After calculating the difficulty of events in $D$, we obtain a new set of prediction orders $\mathbf{o}' = \{o_1', o_2', \ldots, o_{n_e}'\}$. Then, we can predict (the arguments of) events in $D$ from simple to complex according to $\mathbf{o}'$.

The method described above assumes that the model providing the probabilities is well-calibrated, where *confidence* (the probability that a model assigns to a prediction) equals or nearly equals *accuracy* (the real correctness of a prediction). In other words, high confidence corresponds to high accuracy, and vice versa. However, some studies reveal that current Deep Neural Networks (DNNs) are prone to *over-confidence*, which implies that the model's confidence is not reliable (Guo et al.,

2017). We have also found similar problems in our model, which is discussed in Section 5.1. Therefore, we should calibrate these probabilities before using them for our simple-to-complex reordering. Specifically, we adopt temperature scaling (Guo et al., 2017; Desai and Durrett, 2020), a simple and effective method for calibration. In this work, the temperature $T$ is selected by minimizing the Expected Calibration Error (ECE) (Pakdaman Naeini et al., 2015) on the validation set, and we denote the temperature with the lowest ECE as $T'$.

Accounting for calibration, there should be a revision on the calculation of $p_i^{(j)}$. Suppose the logits vector of the $j$-th predicted argument of the $i$-th event be $\mathbf{z}_i^{(j)}$, then:

$$p_i^{(j)} = \max_k softmax \left( \frac{\mathbf{z}_i^{(j)}}{T'} \right)_k, \qquad (8)$$

where $k$ traverses each dimension of $\mathbf{z}_i^{(j)}$.

# 4 Experiments

## 4.1 Dataset and Evaluation Metrics

We evaluate our framework on WIKIEVENTS (Li et al., 2021) as it annotates all the events in a document (averagely 16 events per document), while existing document-level datasets such as DocEE (Tong et al., 2022), RAMS (Ebner et al., 2020) and MUC-4 (Sundheim, 1992) only annotate at most 3 events per document. Also, it provides complete coreference annotation for document-level IAE. Its statistics are shown in Table 1.

| | Train | Dev | Test |
|---|---|---|---|
| # Event Types | 49 | 35 | 34 |
| # Argument Types | 57 | 32 | 44 |
| # Documents | 206 | 20 | 20 |
| # Sentences | 5262 | 378 | 492 |
| # Events | 3241 | 345 | 365 |
| # Arguments | 4413 | 411 | 556 |
| # Events (per doc) | 15.73 | 17.25 | 18.25 |
| # Tokens (per doc) | 789.33 | 643.75 | 712.00 |

Table 1: WIKIEVENTS Statistics

We measure the Argument Identification (Arg-I) and Argument Classification (Arg-C) capabilities of our model following Li et al. (2013). If an argument span matches any of the gold informative arguments of the event, the argument is correctly identified. If the semantic role also matches, the

| Models | Argument Identification (Arg-I) | | | | | | Argument Classification (Arg-C) | | | | | |
|---|---|---|---|---|---|---|---|---|---|---|---|---|
| | Head Match | | | Coref Match | | | Head Match | | | Coref Match | | |
| | P | R | F1 | P | R | F1 | P | R | F1 | P | R | F1 |
| **BERT-CRF** | - | - | $52.71^{\dagger}$ | - | - | $58.12^{\dagger}$ | - | - | $43.29^{\dagger}$ | - | - | $47.70^{\dagger}$ |
| **BART-Gen** | 58.62 | 55.64 | $57.09^{\dagger}$ | 62.84 | 59.64 | $61.19^{\dagger}$ | 54.02 | 51.27 | $52.61^{\dagger}$ | 57.47 | 54.55 | $55.97^{\dagger}$ |
| w/ M | 60.38 | 57.97 | 59.15 | 64.72 | 62.14 | 63.40 | 54.53 | 52.36 | 53.42 | 58.11 | 55.80 | 56.93 |
| w/ M+C | 61.79 | 58.88 | 60.30 | 66.16 | 63.04 | 64.56 | 55.70 | 53.08 | 54.36 | 59.32 | 56.52 | 57.88 |
| w/ M+O (**S2C**) | 61.61 | 59.60 | 60.59 | 65.73 | 63.59 | 64.64 | 55.81 | 53.99 | 54.88 | 59.18 | 57.25 | 58.20 |
| w/ M+O+C' (**S2C-CD**) | 62.57 | 59.96 | $\mathbf{61.24}^{*}$ | 66.73 | 63.95 | $\mathbf{65.31}^{*}$ | 56.52 | 54.17 | $\mathbf{55.32}^{*}$ | 59.92 | 57.43 | $\mathbf{58.65}^{*}$ |

Table 2: Performance (%) of document-level IAE on WIKIEVENTS. M denotes document Memory, C denotes original Constraints from Du et al. (2022), O denotes simple-to-complex reordering, and C' denotes bounded Constraints. $^{\dagger}$ denotes cited results, $^{*}$ denotes statistical significance compared with (Du et al., 2022) ($p < 0.05$).

argument is considered correctly classified. Following previous studies on document-level IAE (Li et al., 2021; Du et al., 2022), we adopt Head Word Match (Head F1) (Huang and Riloff, 2021) and Coreferential Match (Coref F1) (Ji and Grishman, 2008) to judge whether the predicted argument span matches the gold argument span. Head Word Match demands the first word of the predicted argument to match that of the gold argument, while Coreferential Match only needs the extracted argument to be coreferential with the gold argument. We report the micro-P/R/F1 averaged on three different seeds.

## 4.2 Baselines

We compare our framework with a series of competitive baselines: (1) **BERT-CRF** (Shi and Lin, 2019), a simple BERT-based model without incorporating lexical or syntactic features for argument identification and classification. (2) **BART-Gen** (Li et al., 2021), a conditional neural text generation model that generates a filled template for each event given the event template and context words. (3) **BART-Gen (w/ M+C)** (Du et al., 2022), a framework based on BART-Gen, which utilizes retrieval to augment model input and constructs argument pair constraints for decoding. It is the SOTA model on document-level IAE, but still extracts events according to their appearance order in the document. We also report the results of **BART-Gen (w/ M)** for comparison.

## 4.3 Main Results

The main results for document-level IAE are presented in Table 2. From the results, we can conclude that:

- Our S2C-CD model outperforms all previous

methods on WIKIEVENTS as to document-level IAE, with an average gain of 1.4% in F1 on all four settings.

- All models augmented with retrieval (i.e., w/ M) perform better compared with BERT-CRF and raw BART-Gen, showing the importance of modeling inter-event dependencies.

- Compared to BART-Gen (w/ M), the addition of simple-to-complex reordering (S2C model) greatly improves F1, where F1 on average increases by 1.34 in Arg-I and 1.42 in Arg-C. This improvement can be mainly attributed to our simple-to-complex prediction paradigm, since it allows more inter-event dependency links, i.e., from an event to all the events that appeared after it.

- After applying our bounded constraints (S2C-CD model), there is an additional improvement in P and F1, which shows that incorrect constraints are effectively pruned.

## 5 Analysis

## 5.1 Is Calibration Necessary?

**What to Calibrate?** For each argument, we focus on its first token probability, and this probability is what we aim to calibrate. The reason for using the first token probabilities is that the generation of the remaining tokens is highly dependent on the first token. As shown in Figure 2, 87% of the non-first token probabilities are $\geq 0.9$, while first token probabilities are better distributed, with only 53% of them $\geq 0.9$.

**Why to Calibrate?** Modern DNNs are prone to over-confidence, which implies that the model's

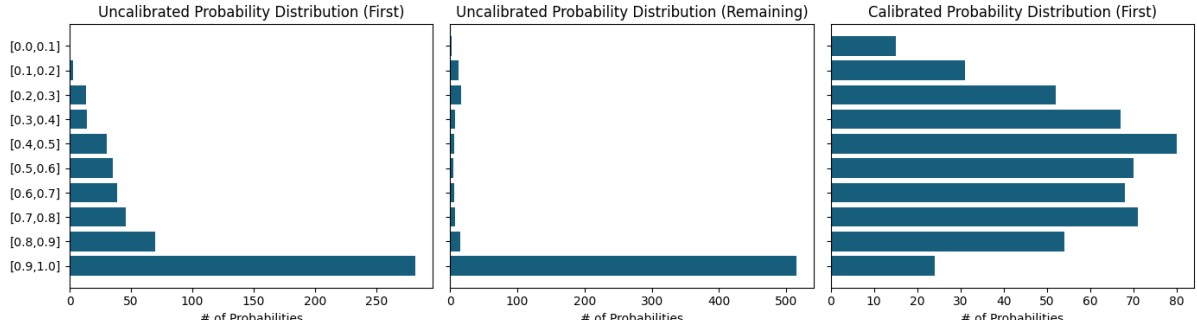

Figure 2: Uncalibrated/Calibrated probability distribution. The left two diagrams respectively show the uncalibrated first and non-first (remaining) token probability distribution, while the diagram on the right shows the calibrated first token probability distribution.

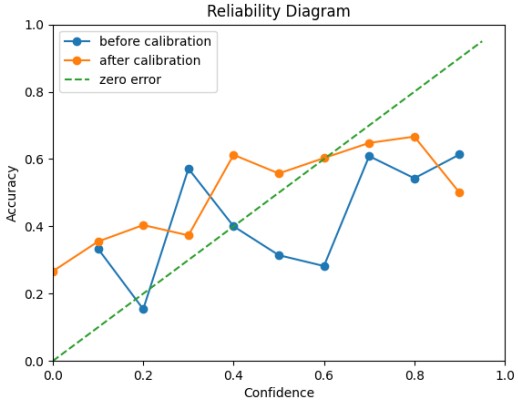

Figure 3: The reliability diagram before/after calibration. The dashed line represents zero error.

| Models | Arg-I | | Arg-C | |
|---|---|---|---|---|
| | Head F1 | Coref F1 | Head F1 | Coref F1 |
| **S2C** | **60.59** | **64.64** | **54.88** | **58.20** |
| – RC | 59.15 | 63.40 | 53.42 | 56.93 |
| + RU | 59.39 | 63.46 | 53.65 | 56.98 |

Table 3: Ablation (%) for simple-to-complex reordering. RC denotes Reordering by Calibrated probabilities (simple-to-complex reordering), RU denotes Reordering by Uncalibrated probabilities.

confidence is not reliable (Guo et al., 2017). We have also found similar problems in our model. As shown in Figure 2, the first and non-first token probability distribution both exhibit a severe over-confidence phenomenon before calibration, with most probabilities $\geq 0.9$. This suggests that the model tends to assign a high (i.e., $\rightarrow 1$) probability to nearly all of the arguments, which can not truly reflect how sure the model is of each argument. Also, over-confidence leads to miscalibration, which is reflected in the *reliability diagram* in Figure 3. The reliability diagram plots the relation between confidence and accuracy, and its definition is discussed in Appendix A.2. As shown in Figure 3, most points on the orange curve (before calibration) are far from the zero error curve where confidence exactly equals accuracy, demonstrating that uncalibrated probabilities are unreliable. After calibration, the first token probability distribution becomes flat (Figure 2) and calibrated (Figure 3). Therefore, we should calibrate probabilities (con-

fidence) to align them with accuracy before using them for our simple-to-complex reordering.

**Influence of Uncalibrated Probabilities** We conduct an ablation study to further explore what will happen if we reorder events using uncalibrated probabilities. As shown in Table 3, if we order events by uncalibrated probabilities, F1 is only comparable to excluding simple-to-complex reordering from our S2C model. The performance is maximized only when the probabilities are calibrated. Therefore, calibration is essential.

## 5.2 Difficulty Calculation Needs Memory?

In this section, we present two possible ways of calculating the difficulty of an event to explore their impact on performance. In Figure 4, we define the **first inference** as step 1-2 (calculating the difficulty), and the **second inference** as step 3 (predicting the arguments of reordered events).

The first way is utilized in our framework, where we use the same model for both inferences. When calculating the difficulty at the first inference, we also use retrieval. There are two reasons for this. On the one hand, the model input is augmented with retrieval during training, so the input/prompt

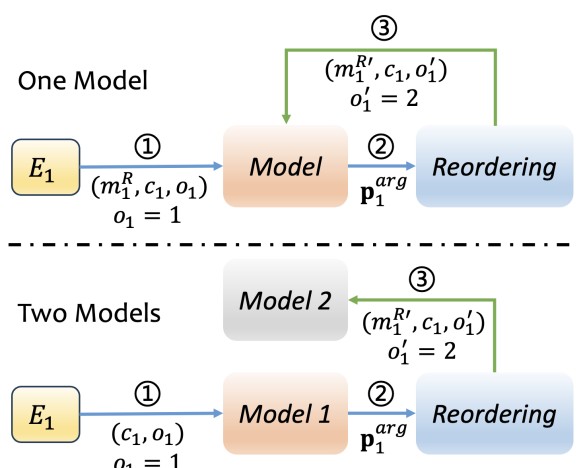

Figure 4: Two ways of calculating the difficulty.

| Intervals | Arg-I | | Arg-C | |
|---|---|---|---|---|
| | Head F1 | Coref F1 | Head F1 | Coref F1 |
| **S2C** (R1) | 60.59 | 64.64 | **54.88** | **58.20** |
| **S2C** (R2) | **60.79** | 64.65 | 54.55 | 57.67 |

Table 4: Performance (%) of using two different ways to calculate difficulty. R1 and R2 denote the results of the first and second way, respectively.

format should be consistent during testing. On the other hand, the model is trained to predict the arguments of an upcoming event conditioned on the prediction of an already predicted event. Therefore, we should calculate "the difficulty of an event conditioned on the prediction of an earlier predicted event". However, the retrieved predictions of the same event at both inferences are usually different.

The second way is to use separate models for each inference. At the first inference, we use a model trained without retrieval to calculate the "raw" difficulty of an event (i.e., do not condition on the prediction of an earlier predicted event). At the second inference, we train a retrieval-augmented model. This way removes the possible influence of retrieval on calculating the difficulty of an event, but the training overhead doubles.

We conduct an experiment to compare these two ways, as shown in Table 4. R1 and R2 represent one model and two models, respectively. R1 is comparable to R2 in Arg-I, while R1 is notably better than R2 in Arg-C. The results suggest that using one model is generally better as to performance, so we should calculate the difficulty of an event conditioned on the prediction of an earlier predicted event. Besides, using one model is more time-efficient.

### 5.3 Influence of Bounded Constraints

In this section, we first compare our bounded constraints with those presented in Du et al. (2022), then analyze the impact of the lower and upper bounds individually.

In Table 5, we observe that when applying the original constraints (Du et al., 2022), the model

performs only comparably with our S2C model. This implies the number of correct and incorrect constraints is nearly equal when we predict events from simple to complex. By contrast, our bounded constraints perform well, suggesting that the number of incorrect constraints is indeed reduced and the correct constraints are (mostly) kept.

| Models | Arg-I | | Arg-C | |
|---|---|---|---|---|
| | Head F1 | Coref F1 | Head F1 | Coref F1 |
| **S2C-CD** | **61.24** | **65.31** | **55.32** | **58.65** |
| − BCs (**S2C**) | 60.59 | 64.64 | 54.88 | 58.20 |
| + OCs | 60.61 | 64.69 | 54.87 | 58.20 |

Table 5: Ablation (%) for bounded constraints. BCs denote Bounded Constraints, OCs denote Original Constraints from Du et al. (2022).

Using the steps presented in Appendix A.3, we obtain the lower bound $0.5$ and the upper bound $0.8$, so we will disable a constraint if the probability of decoding the argument is $\leq 0.5$ or $\geq 0.8$. Based on this, we individually analyze the influence of the lower and upper bounds, as shown in Table 6. We have found that whether we remove the lower bound or the upper bound, the performance drops, indicating that both bounds are useful for reducing the number of incorrect constraints.

| Intervals | Arg-I | | Arg-C | |
|---|---|---|---|---|
| | Head F1 | Coref F1 | Head F1 | Coref F1 |
| $[0.5, 0.8]$ | **61.24** | **65.31** | **55.32** | **58.65** |
| $[0.0, 0.8]$ | 60.72 | 64.81 | 54.78 | 58.12 |
| $[0.5, 1.0]$ | 60.56 | 64.63 | 54.81 | 58.15 |

Table 6: Influence of the lower and upper bounds.

### 5.4 Case Study

In the case presented in Figure 5, we would like to predict the arguments of event $E = Damage$, which describes the mental damage that Dzhokhar Tsarnaev brought to the victims as a bomber. In

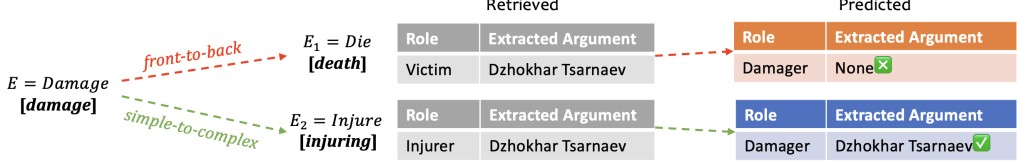

[S1] Moments before a federal judge sentenced him to **death**, Boston Marathon bomber [Dzhokhar Tsarnaev] rose to his feet Wednesday and apologized to the victims and their loved ones for the first time.

[S3] Amid deep silence in the courtroom, the 21-year-old ethnic Chechen said, "I am sorry for the lives that I've taken, for the suffering that I've caused you, for the **damage** that I've done — irreparable damage."

[S7] In May, after a 12-week trial, Tsarnaev was found guilty of killing three people and **injuring** 264 in the April 15, 2013, bombing at the world-renowned race, where he and his brother, Tamerlan, 26, set off two pressure-cooker bombs near the finish line.

Figure 5: Case study on simple-to-complex reordering.

front-to-back prediction, $E$ can only access the predictions of earlier appeared events and retrieves $E_1$'s prediction as additional input. The death of Dzhokhar Tsarnaev in $E_1$ happened after $E$, wrongly restricting the prediction of "Dzhokhar Tsarnaev" as the DAMAGER of $E$. By contrast, with our simple-to-complex prediction, $E$ has the chance to rely on the INJURER argument of a later appeared event $E_2$ and obtain the correct DAMAGER argument "Dzhokhar Tsarnaev". $E_2$ describes that Dzhokhar Tsarnaev injured 264 people in the bombing as the INJURER, thus bringing mental damage to them as the DAMAGER.

Comparing these two prediction paradigms, we find that simple-to-complex prediction is better, mainly because it allows more inter-event dependency links. In this example, it is intuitive that $E$ is more similar to $E_2$, since they respectively depict the physical damage and mental damage Dzhokhar Tsarnaev brought to the victims. However, the dependency link from $E$ to $E_2$ is disabled when predicting events from front to back.

## 5.5 Error Analysis

Table 7 summarizes the error types of our S2C-CD model. The errors mainly come from the inability to recognize an argument span (around half), while only about 8% of identified arguments are assigned incorrect semantic roles. Therefore, identifying the argument span is more important than assigning a more accurate role to already extracted arguments.

## 6 Related Work

### 6.1 Document-level EAE

Unlike sentence-level EAE (Li et al., 2014; Du and Cardie, 2020; Xiangyu et al., 2021), events

|  | **Unidentified** | **Spurious** | **Misclassified** |
|---|---|---|---|
| HM | 221 (49.0%) | 198 (43.9%) | 32 (7.1%) |
| CM | 199 (48.4%) | 176 (42.8%) | 36 (8.8%) |

Table 7: Errors made by our framework under Head Match (HM) and Coref Match (CM).

and their participants usually spread across the document in document-level EAE. We focus on document-level IAE (Li et al., 2021) in this work, which is more practical but more challenging. Li et al. (2021) constructed the WIKIEVENTS dataset and pioneered the research on document-level IAE. Compared with existing document-level EAE datasets such as DocEE (Tong et al., 2022), RAMS (Ebner et al., 2020) and MUC-4 (Sundheim, 1992) that only annotate at most 3 events per document, WIKIEVENTS annotates all the events in a document, with an average of 16 events per document. Also, it provides complete coreference annotation for evaluating document-level IAE.

Recently, generation-based methods have been proposed for document-level EAE. Among them, template generation-based approaches (Li et al., 2021; Huang et al., 2022; Du et al., 2022) are widely utilized. BART-Gen (Li et al., 2021) conditioned generation on event templates and context words but considered each event independently. Further, Du et al. (2022); Du and Ji (2022) introduced the idea of "memory" to document-level EAE, where predictions of already predicted events were utilized as additional input. Although these methods can model inter-event dependencies to some extent, they ignore the dependency links from an event to all the events that appeared after it in a document. Besides, uncertain/false event predic-

tions may be cached first and retrieved by future events, misleading their prediction.

## 6.2 Confidence Calibration

Studies on the calibration of natural language models have been drawing attention recently (Desai and Durrett, 2020; Park and Caragea, 2022; Kim et al., 2023). Among modern calibration approaches, temperature scaling is a simple and effective method (Desai and Durrett, 2020) which can produce low ECE (Guo et al., 2017; Chen et al., 2023). Due to its low time overhead and low ECE property, we adopt it in our work. Other works focus on methods such as label smoothing (Pereyra et al., 2017) and data augmentation (Hendrycks* et al., 2020), but these methods cannot produce as low ECE as temperature scaling (Chen et al., 2023). More recent studies started to treat calibration as an additional task, which needs collecting data and training extra models (Ye and Durrett, 2022; Zhang et al., 2021). In order to reduce time overhead, we do not consider these approaches.

## 7 Conclusion

In this work, we propose the idea of *simple-to-complex* prediction for events in a document, where events in a document are reordered from simple to complex and predicted accordingly. Besides, we introduce retrieval to augment model input and apply constrained decoding to improve model output. Empirical results and analysis demonstrate that our best model outperforms prior methods by a notable margin and our simple-to-complex prediction is beneficial since it allows more inter-event dependency links, i.e., from an event to all the events appeared after it.

## Limitations

Firstly, our framework requires two inference processes, where the first inference is to calculate the difficulty of events in a document and the second inference is to predict the arguments of these events from simple to complex. Secondly, the way of setting lower/upper bounds is a hard pruning strategy that disables constraints where the argument probability is too low/high. However, this strategy rigidly excludes constraints for which the model is not sufficiently certain or less reliable, without really taking into account the wrong constraints caused by the incorrectly predicted arguments. We leave the problems for future work.

## Acknowledgements

This work is supported by the National Key R&D Program of China (No. 2021YFC3340304). We would like to thank the anonymous reviewers for their helpful comments. We would like to express appreciation to Yansong Feng for his insightful suggestions on our idea.

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

# A Appendix

## A.1 Hyperparameters

The hyperparameters used in our experiments are shown in Table 8.

| Hyperparameter | Value |
| --- | --- |
| train_batch_size | 2 |
| eval_batch_size | 1 |
| learning_rate | 3e-5 |
| accumulate_grad_batches | 4 |
| train_epoches | 5 |
| warmup_steps | 0 |
| weight_decay | 0 |
| # gpus | 1 |

Table 8: Hyperparameters.

## A.2 Confidence Calibration

**Confidence & Accuracy**   Confidence is defined as the probability that a model assigns to a prediction, while accuracy is the real correctness of a prediction. In the classification task, "prediction" means the predicted class of a specific instance. In our work, "prediction" means a specific argument.

**Confidence Calibration**   The goal of confidence calibration is to align the model's posterior probabilities (confidence) with empirical likelihoods (accuracy) (Guo et al., 2017). For example, if we take 100 instances where the model's prediction receives a posterior probability of 0.8, the model should get 80 of the instances correct.

**Reliability Diagrams**   Usually, calibration is visualized by reliability diagrams (Degroot and Fienberg, 1983; Niculescu-Mizil and Caruana, 2005). Reliability diagrams treat expected accuracy as a function of model confidence. To draw the reliability diagram, we usually partition predictions into $k$ disjoint, equally-sized bins $\{B_1, B_2, \ldots, B_k\}$, and calculate the average confidence/accuracy in each bin as an approximation. In this paper, we set $k = 10$.

**Evaluation Metrics**   We use Expected Calibration Error (ECE) (Pakdaman Naeini et al., 2015) in this work. ECE is the weighted average of all the bins' confidence/accuracy difference in the reliability diagram:

$$\text{ECE} = \sum_{i=1}^{k} \frac{|B_i|}{n} |\text{acc}(B_i) - \text{conf}(B_i)| \quad (9)$$

where $n$ denotes the number of predictions. ECE measures how calibrated the model is. The smaller the ECE, the more calibrated the model is.

## A.3 The Heuristics of Selecting Bounds

We select the lower and upper bounds of the argument probabilities according to the probability distribution (Figure 2) and the reliability diagram (Figure 3). The steps are as follows:

1. Firstly, calculate the median $N_{med}$ of the number of probabilities in each interval (e.g., $[0.9, 1.0]$) of the calibrated probability distribution (Figure 2). If the number of probabilities in an interval is $\geq N_{med}$, then it is selected as a candidate interval.

2. Secondly, merge all candidate intervals to form a continuous interval $I$. If there are multiple intervals, then prune intervals with poorer calibration (e.g., $[0.8, 1.0]$) according to the reliability diagram (Figure 3), and only keep one (denoted as $I$).

3. Finally, prune the less calibrated part of $I$.

Following these steps, we present the process of selecting bounds in our work below. First, we can calculate $N_{med} = 60.5$ according to Figure 2. After merging candidate intervals, we obtain a continuous interval $I = [0.3, 0.8]$. Then, we should prune $[0.3, 0.5]$ because probabilities in this interval are less calibrated with low accuracy (Figure 3). To sum up, our final interval is $[0.5, 0.8]$.