# OpenReview forum: "From Simple to Complex: A Progressive Framework for Document-level Informative Argument Extraction"
_EMNLP/2023/Conference — EMNLP 2023 Findings_

### Official Review · Reviewer_CXkm · 2023-07-31

**Soundness:** 3

**Excitement:**

3: Ambivalent: It has merits (e.g., it reports state-of-the-art results, the idea is nice), but there are key weaknesses (e.g., it describes incremental work), and it can significantly benefit from another round of revision. However, I won't object to accepting it if my co-reviewers champion it.

**Paper Topic And Main Contributions:**

This paper proposes a new method for document-level informative argument extraction via a simple-to-complex progressive framework. The proposed order of the predicted events in a document by their difficulties to eliminate the error propagation of argument extraction.

**Questions For The Authors:**

1. Is it reasonable to use similarity to retrieve predicted event templates and arguments to predict probabilities to define the difficulty of event extraction?

**Reasons To Accept:**

1. This paper identifies and locates the error propagation problem of sequential prediction in document-level event argument detection. And trying to distinguish the difficulty of the event to alleviate the error propagation, which has certain reference significance.

**Reasons To Reject:**

1. The experimental performance improvement of the proposed model is not obvious, and some significance tests can be added to verify whether the improved effect is credible.
2. Some core modules are not clearly written, especially the Order Adaptive Argument Pair Constraints part. It is suggested to add examples or descriptions as necessary.

**Reproducibility:**

5: Could easily reproduce the results.

**Reviewer Confidence:**

4: Quite sure. I tried to check the important points carefully. It's unlikely, though conceivable, that I missed something that should affect my ratings.

---

> ### Author Rebuttal · Authors · 2023-08-28
>
> Thank you for your comments! We will answer your questions (both in Reasons To Reject and Questions For The Authors) about our work one by one as follows:
>
> **Q1:** The experimental performance improvement of the proposed model is not obvious, and some significance tests can be added to verify whether the improved effect is credible.
>
> **A1:** Thanks for your suggestions. We conduct our experiments three times for each setting and report the average micro-F1 in our paper. We will add the variance and $p$-value in the final version, which are provided below. The $p$-value is calculated between our best model and the previous SOTA (both marked in bold in the table below). As is shown in the table, all the $p$-values are less than 0.05, indicating our improvement is significant.
>
> | Models                              | Arg-I Head F1 | Arg-I Coref F1 | Arg-C Head F1 | Arg-C Coref F1 |
> | ----------------------------------- | :-----------: | :------------: | :-----------: | :------------: |
> | BART-Gen w/ M                       |  59.15±0.27   |   63.40±0.35   |  53.42±0.34   |   56.93±0.38   |
> | **BART-Gen w/ M+C (Previous SOTA)** |  60.30±0.11   |   64.56±0.31   |  54.36±0.25   |   57.88±0.34   |
> | BART-Gen w/ M+O (Our Model)         |  60.59±0.23   |   64.64±0.21   |  54.88±0.26   |   58.20±0.39   |
> | **BART-Gen w/ M+O+C’ (Our Model)**  |  61.24±0.14   |   65.31±0.27   |  55.32±0.20   |   58.65±0.20   |
> | $p$-value                           |  0.01088625   |   0.00211084   |  0.00127592   |   0.00825234   |
>
> **Q2:** Some core modules are not clearly written, especially the Order Adaptive Argument Pair Constraints part. It is suggested to add examples or descriptions as necessary.
>
> **A2:** Thanks for your suggestions, we will include more detailed information as well as examples about this module. % Below are an introduction of this module?
>
> The argument pair constraints are used at inference time to constrain the prediction of conflicting entities in a document. For example, if the "Detainee" of event $e_1$ is "Mike", then "Mike" cannot be predicted as the "Attacker" of another event $e_2$ (happened after $e_1$) in the same document, since "Mike" is already in jail. After "Mike" is decoded in $e_1$ as "Detainee", we lower the probability of predicting it as the "Attacker" of another event $e_2$ to restrict such wrong/contradictory prediction.
>
> However, this kind of constraints assume that previously predicted arguments are all correct and can be used to constrain the prediction of next-event arguments, which cannot be true. Once an incorrect prediction is used to constrain another, it may cause more errors sometimes. In the example above, "Mike" will never be predicted as the "Attacker" of $e_2$ once he is predicted as the "Detainee" of $e_1$, even if the prediction of "Detainee" is incorrect. To alleviate such error propagation, we only apply the constraints when the model is certain about the prediction of an entity, where the certainty of an entity can be measured by its output probability when decoding. When the output probability of an entity (e.g., "Mike") is too low/high, we disable the constraints. Low probability means the model is not sure/certain about this prediction, while high probability implies the prediction is not reliable. As is shown in Figure 4, when the probability is high (e.g., *>*0.8), the accuracy is low. Therefore, these predictions are not reliable, and we should prune them.
>
> **Q3:** Is it reasonable to use similarity to retrieve predicted event templates and arguments to predict probabilities to define the difficulty of event extraction?
>
> **A3:** We think, the question is why do we retrieve event templates and arguments from the document memory as additional input when predicting probabilities as the measure of difficulty?
>
> The reason is that when calculating the difficulty (first inference) and predicting the final results (second inference), we use the same model, and its objective is to predict the next event conditioned on the retrieved event templates/arguments. In other words, the input/prompt format consistency should be ensured. However, we could also exclude retrieval when predicting the difficulty to eliminate its possible influence in the calculation process. Specifically, we use a separate BART model trained without retrieval to predict probabilities, and use the originally trained BART model (with retrieval) to get the final results. R1 and R2 are the results presented in the paper, and R3 is the result of this experiment. Comparing R2 and R3, we can conclude that the exclusion of retrieval does not have a large impact on model performance, where R2 is even slightly better (especially in Arg-C) than R3, proving that it is reasonable to include retrieval when calculating the difficulty.
>
> | Models                   | Arg-I Head F1  | Arg-I Coref F1 | Arg-C Head F1  | Arg-C Coref F1 |
> | ------------------------ | :------------: | :------------: | :------------: | :------------: |
> | BART-Gen w/ M **(R1)**   |   59.15±0.27   |   63.40±0.35   |   53.42±0.34   |   56.93±0.38   |
> | BART-Gen w/ M+O **(R2)** |   60.59±0.23   |   64.64±0.21   | **54.88**±0.26 | **58.20**±0.39 |
> | BART-Gen w/ M+O **(R3)** | **60.79**±0.23 | **64.65**±0.24 |   54.55±0.22   |   57.67±0.28   |

---

### Official Review · Reviewer_FLW2 · 2023-08-05

**Soundness:** 3

**Excitement:**

3: Ambivalent: It has merits (e.g., it reports state-of-the-art results, the idea is nice), but there are key weaknesses (e.g., it describes incremental work), and it can significantly benefit from another round of revision. However, I won't object to accepting it if my co-reviewers champion it.

**Paper Topic And Main Contributions:**

The paper investigates the task of document level event argument extraction: Given a set of event triggers in a document, the task is to identify their corresponding arguments. The paper improves upon an existing generative method for task, by ordering the events to be processed by the observed probability of the generated arguments rather than in order of appearance.

The suggested method is evaluated in an empirical study and results suggest that it slightly improves upon the state of the art on an existing wikipedia-based event argument extraction dataset.

**Questions For The Authors:**

Question A: What is the train/dev/test size?

Question B: How were the P/R/F1 averaged? Micro, Macro, Instance-based?

Question C: Did you conduct any statistical analysis of the results? Are the runs averaged over multiple seeds?

I am open to pushing my soundness score if I am convinced that the results are statistically sound

**Reasons To Accept:**

The proposed method is simple and is motivated and explained well, it also seems to be effective. The empirical analysis is reasonably thorough and attributes the performance gains to the proposed improvements over the baseline method.

**Reasons To Reject:**

The improvements stemming from the proposed method over the baseline are only marginal. This in itself is not a problem, however the difference should be analysed with statistical tests, to inspire confidence that the improvements are indeed due to the proposed method rather than chance.

Some minor details regarding the experimental setup are missing (see my questions to the authors).

Finally, the contributions here might be considered too small to warrant acceptance at a flagship conference: The proposed method is an incremental improvement, the score improvements are marginal, the task is rather niche with only 1 dataset available. As such I am not too excited about this submission. However, I think the paper deserves its place in the Findings volume.

**Reproducibility:**

4: Could mostly reproduce the results, but there may be some variation because of sample variance or minor variations in their interpretation of the protocol or method.

**Reviewer Confidence:**

4: Quite sure. I tried to check the important points carefully. It's unlikely, though conceivable, that I missed something that should affect my ratings.

**Typos Grammar Style And Presentation Improvements:**

There are some slight grammatical errors, e.g. ll048--052 "Recent works model [...] and achieves [...]"

The manuscript could benefit from a quick run through a grammar checker (e.g. grammarly or Docs/Word) to get rid of those.

---

> ### Author Rebuttal · Authors · 2023-08-28
>
> Thank you for your enlightening questions and suggestions! Firstly, we will respond to your questions about the contribution of our work in **Reasons To Reject**.
>
> Although we only conduct experiments on WikiEvents, this does not mean that our work is merely incremental. That is because we made a significant improvement on a very challenging task. Our proposed method aims to solve the poblems where there are multiple events in one document and the arguments might be very distant from the trigger. It is a very challenging task and to the best of our knowledge, WikiEvents is the only dataset suitable for this task. Due to the difficulty of annotation, other datasets like DocEE$^{[1]}$, RAMS$^{[2]}$ and MUC-4$^{[3]}$ only annotate one or two events per document and they mostly focus on local extraction, where arguments are very close to the trigger. Instead, WikiEvents has 16 events per document averagely, and focus on the extraction of informative mentions, which are 14*×* distant from the trigger than their uninformative/local counterparts. Therefore, the setting of WikiEvents is suitable for our task, and our experiments on this dataset prove the effectiveness of our framework. Since this task is more challenging compared to traditional document-level EAE, it is less explored with only a few works$^{[4-5]}$. Besides, the improvement of our framework over previous SOTA is statistically significant, and we will provide the results when replying to your Question C.
>
> ---
>
> We have also read your **Questions for The Authors** carefully, and would like to answer them one by one as follows:
>
> **Question A:** What is the train/dev/test size?
>
> **Answer:** WikiEvents statistics are shown in Table 1, where there are 3241, 345, and 365 events in the train/dev/test set, respectively. Besides, there are respectively 4413, 411, and 556 arguments in the train/dev/test set, which is not presented in the paper. We will include this in our final version.
>
> **Question B:** How were the P/R/F1 averaged? Micro, Macro, Instance-based?
>
> **Answer:** We follow previous works$^{[4-5]}$ and calculate micro-P/R/F1 for a fair comparison. In the setting of EAE, this calculation is argument-level (not event/instance-level), where we first calculate P and R for all the arguments, then calculate a general F1.
>
> **Question C:** Did you conduct any statistical analysis of the results? Are the runs averaged over multiple seeds?
>
> **Answer:** Yes, we conduct our experiments three times on three different seeds for each setting and report the average micro-F1 in our paper. We will add the variance and $p$-value in the final version, which are provided below. The $p$-value is calculated between our best model and the previous SOTA (both marked in bold in the table below). As is shown in the table, all the $p$-values are less than 0.05, indicating our improvement is significant. Besides, our best model outperforms the previous SOTA on each seed.
>
> | Models                              | Arg-I Head F1 | Arg-I Coref F1 | Arg-C Head F1 | Arg-C Coref F1 |
> | ----------------------------------- | :-----------: | :------------: | :-----------: | :------------: |
> | BART-Gen w/ M                       |  59.15±0.27   |   63.40±0.35   |  53.42±0.34   |   56.93±0.38   |
> | **BART-Gen w/ M+C (Previous SOTA)** |  60.30±0.11   |   64.56±0.31   |  54.36±0.25   |   57.88±0.34   |
> | BART-Gen w/ M+O (Our Model)         |  60.59±0.23   |   64.64±0.21   |  54.88±0.26   |   58.20±0.39   |
> | **BART-Gen w/ M+O+C’ (Our Model)**  |  61.24±0.14   |   65.31±0.27   |  55.32±0.20   |   58.65±0.20   |
> | $p$-value                           |  0.01088625   |   0.00211084   |  0.00127592   |   0.00825234   |
>
> [1] Tong, M., Xu, B., Wang, S., Han, M., Cao, Y., Zhu, J., ... & Li, J. (2022, July). DocEE: A Large-Scale and Fine-grained Benchmark for Document-level Event Extraction. In *Proceedings of the 2022 Conference of the North American Chapter of the Association for Computational Linguistics: Human Language Technologies* (pp. 3970-3982).
>
> [2] McLean, V. (1992, June). Fourth message understanding conference (MUC-4). In *Proceedings of fourth message understanding conference (MUC-4)*.
>
> [3] Ebner, S., Xia, P., Culkin, R., Rawlins, K., & Van Durme, B. (2020, July). Multi-Sentence Argument Linking. In *Proceedings of the 58th Annual Meeting of the Association for Computational Linguistics* (pp. 8057-8077).
>
> [4] Li, S., Ji, H., & Han, J. (2021, June). Document-Level Event Argument Extraction by Conditional Generation. In *Proceedings of the 2021 Conference of the North American Chapter of the Association for Computational Linguistics: Human Language Technologies* (pp. 894-908).
>
> [5] Du, X., Li, S., & Ji, H. (2022, May). Dynamic Global Memory for Document-level Argument Extraction. In *Proceedings of the 60th Annual Meeting of the Association for Computational Linguistics (Volume 1: Long Papers)* (pp. 5264-5275).

---

### Official Review · Reviewer_Unoe · 2023-08-19

**Soundness:** 4

**Excitement:**

4: Strong: This paper deepens the understanding of some phenomenon or lowers the barriers to an existing research direction.

**Paper Topic And Main Contributions:**

This paper is about document-level event argument extraction task.
The authors argue that extracting events according to their appearance order has ignored the difficulty of each event. They propose a simple-to-complex progressive framework for document-level informative argument extraction.
Experiment results have shown that their proposed method can help event arguments prediction and have achieved outperform performance.

**Questions For The Authors:**

A. The equipment and environment of your experiment? How long does it take for each runs?

B. Are there cases of multiple labels? E.g. an argument role is annotated with two or more entities? Or an entity plays more than one argument role in event? How you deal with that?

C. Are all results averaged over multiple seeds?

**Reasons To Accept:**

This paper proposes a novel simple-to-complex progressive framework for document-level event argument extraction. The paper is well written and clearly organized. The authors conduct comprehensive experiments to validate their argument about simple-to-complex framework. They also experiment and analysis the prediction probabilities bias issues and the entity type constraints issues.

**Reasons To Reject:**

A. This paper only conducts experiments on WikiEvents dataset and compares the results with three previous works, which may be insufficient.

B. Fig.5 only presents the results that the model can predict correctly with simple-to-complex ordering, but if without it cannot. I would prefer to see how the simple-to-complex ordering corrects the prediction, in this case study.

C. The pipeline method of two step inference processes also brings some error propagattions for the final event arguments extraction.

D. The introduction section should include a brief description of the proposed methods.

**Reproducibility:**

4: Could mostly reproduce the results, but there may be some variation because of sample variance or minor variations in their interpretation of the protocol or method.

**Reviewer Confidence:**

3: Pretty sure, but there's a chance I missed something. Although I have a good feel for this area in general, I did not carefully check the paper's details, e.g., the math, experimental design, or novelty.

---

> ### Author Rebuttal · Authors · 2023-08-28
>
> Thanks for your thoughtful review comments! Firstly, I will reply to the points that you have mentioned in **Reasons To Reject**.
>
> **Reason A:** This paper only conducts experiments on WikiEvents dataset and compares the results with three previous works, which may be insufficient.
>
> **Response:** Our proposed method aims to solve the poblems where there are multiple events in one document and the arguments might be very distant from the trigger. It is a very challenging task and to the best of our knowledge, WikiEvents is the only dataset suitable for this task. Due to the difficulty of annotation, other datasets like DocEE$^{[1]}$, RAMS$^{[2]}$ and MUC-4$^{[3]}$ only annotate one or two events per document and they mostly focus on local extraction, where arguments are very close to the trigger. Instead, WikiEvents has 16 events per document averagely, and focus on the extraction of informative mentions, which are 14*×* distant from the trigger than their uninformative/local counterparts. Therefore, the setting of WikiEvents is suitable for our task, and our experiments on this dataset prove the effectiveness of our framework. Since this task is more challenging compared to traditional document-level EAE, it is less explored with only a few works$^{[4-5]}$.
>
> **Reason B:** Fig.5 only presents the results that the model can predict correctly with simple-to-complex ordering, but if without it cannot. I would prefer to see how the simple-to-complex ordering corrects the prediction, in this case study.
>
> **Response:** In this case, the order of difficulty in predicting the three events is $E_2>E>E_1$. The prediction order before ordering is $E_1\rightarrow E_2\rightarrow E$, where $E_2$ is the most difficult event with the low probability and is retrieved as additional input for $E$, misleading the prediction. After ordering, the prediction order is $E_1\rightarrow E\rightarrow E_2$, where $E_1$ provides a reliable informative argument "Dzhokhar Tsarnaev" to help the prediction of $E$, and $E_2$ is excluded because it is more difficult than $E$.
>
> **Reason C:** The pipeline method of two step inference processes also brings some error propagattions for the final event arguments extraction.
>
> **Response:** Any pipeline will have the error propagation problem. In this work, the quality of ordering determines the final result. Although ordering may bring errors sometimes, the advantages still outweigh the disadvantages, as shown in the table below. This table summarizes the prediction change from BART-Gen (w/ M) to BART-Gen (w/ M+O) on the test set. There are in total 12 cases that increase F1, while only 3 cases harm F1.
>
> | Prediction chang        | Number of cases |
> | ----------------------- | --------------- |
> | **Wrong->Right (good)** | 12              |
> | **Right->Wrong (bad)**  | 3               |
>
> **Reason D:** The introduction section should include a brief description of the proposed methods.
>
> **Response:** Thanks for your suggestion. We will add a brief description of our framework to the introduction section in the final version.
>
> ---
>
> Secondly, we would like to respond to your **Questions for The Authors** one by one as follows:
>
> **Question A:** The equipment and environment of your experiment? How long does it take for each runs?
>
> **Answer:** We train and test our model using a single NVIDIA GeForce RTX 3090 GPU. It takes about 550 seconds for training and 320 seconds for inference. There may also be some checkpoint loading overhead, but it's not a high percentage compared to the total time consumption.
>
> **Question B:** Are there cases of multiple labels? E.g. an argument role is annotated with two or more entities? Or an entity plays more than one argument role in event? How you deal with that?
>
> **Answer:** Our method is generation-based, so it is flexible and natural to handle these two situations. If an argument role is annotated with two (or more) entities, the model can predict as "entity_A and entity_B" in the corresponding slot of the event template. If an entity plays multiple roles in a specific event, the same entity can be predicted in two (or more) slots of the event template, since generative approaches do not restrict one entity to be decoded only once.
>
> **Question C:** Are all results averaged over multiple seeds?
>
> **Answer:** Yes, we conduct our experiments three times on three different seeds for each setting and report the average micro-F1 in our paper. We will add the variance and $p$-value in the final version, which are provided below. The $p$-value is calculated between our best model and the previous SOTA (both marked in bold in the table below). As is shown in the table, all the $p$-values are less than 0.05, indicating our improvement is significant.
>
> | Models                              | Arg-I Head F1 | Arg-I Coref F1 | Arg-C Head F1 | Arg-C Coref F1 |
> | ----------------------------------- | :-----------: | :------------: | :-----------: | :------------: |
> | BART-Gen w/ M                       |  59.15±0.27   |   63.40±0.35   |  53.42±0.34   |   56.93±0.38   |
> | **BART-Gen w/ M+C (Previous SOTA)** |  60.30±0.11   |   64.56±0.31   |  54.36±0.25   |   57.88±0.34   |
> | BART-Gen w/ M+O (Our Model)         |  60.59±0.23   |   64.64±0.21   |  54.88±0.26   |   58.20±0.39   |
> | **BART-Gen w/ M+O+C’ (Our Model)**  |  61.24±0.14   |   65.31±0.27   |  55.32±0.20   |   58.65±0.20   |
> | $p$-value                           |  0.01088625   |   0.00211084   |  0.00127592   |   0.00825234   |
>
> [1] Tong, M., Xu, B., Wang, S., Han, M., Cao, Y., Zhu, J., ... & Li, J. (2022, July). DocEE: A Large-Scale and Fine-grained Benchmark for Document-level Event Extraction. In *Proceedings of the 2022 Conference of the North American Chapter of the Association for Computational Linguistics: Human Language Technologies* (pp. 3970-3982).
>
> [2] Ebner, S., Xia, P., Culkin, R., Rawlins, K., & Van Durme, B. (2020, July). Multi-Sentence Argument Linking. In *Proceedings of the 58th Annual Meeting of the Association for Computational Linguistics* (pp. 8057-8077).
>
> [3] McLean, V. (1992, June). Fourth message understanding conference (MUC-4). In *Proceedings of fourth message understanding conference (MUC-4)*.
>
> [4] Li, S., Ji, H., & Han, J. (2021, June). Document-Level Event Argument Extraction by Conditional Generation. In *Proceedings of the 2021 Conference of the North American Chapter of the Association for Computational Linguistics: Human Language Technologies* (pp. 894-908).
>
> [5] Du, X., Li, S., & Ji, H. (2022, May). Dynamic Global Memory for Document-level Argument Extraction. In *Proceedings of the 60th Annual Meeting of the Association for Computational Linguistics (Volume 1: Long Papers)* (pp. 5264-5275).

---

### Meta-Review · Area_Chair_kqfZ · 2023-09-19

**Recommendation:** 3

**Metareview:**

This paper presents a simple-to-complex progressive approach for event argument extraction and the complexities of events are determined by the probability of their generated arguments in the first step. The approach achieves better extraction performance but the improvements are relatively marginal.

---

### Decision · Program_Chairs · 2023-10-07

**Decision:**

Accept-Findings

**Comment:**

This paper presents a simple-to-complex progressive approach for event argument extraction and the complexities of events are determined by the probability of their generated arguments in the first step. The approach achieves better extraction performance but the improvements are relatively marginal.